# Safeguarding against Dementia in Aboriginal and Torres Strait Islander Communities through the Optimisation of Primary Health Care: A Project Protocol

**DOI:** 10.3390/mps6050103

**Published:** 2023-10-19

**Authors:** Yvonne C. Hornby-Turner, Sarah G. Russell, Rachel Quigley, Veronica Matthews, Sarah Larkins, Noel Hayman, Prabha Lakhan, Leon Flicker, Kate Smith, Dallas McKeown, Diane Cadet-James, Alan Cass, Gail Garvey, Dina LoGiudice, Gavin Miller, Edward Strivens

**Affiliations:** 1College of Medicine and Dentistry, James Cook University, Smithfield, QLD 4878, Australia; sarah.russell6@jcu.edu.au (S.G.R.); rachel.quigley@jcu.edu.au (R.Q.); edward.strivens@health.qld.gov.au (E.S.); 2University Centre for Rural Health, University of Sydney, Lismore, NSW 2145, Australia; 3Southern Queensland Centre of Excellence in Aboriginal and Torres Strait Islander Primary Health Care, Metro South Hospital and Health Service, Inala, QLD 4077, Australiaprabha.lakhan@health.qld.gov.au (P.L.); 4Western Australian Centre for Health and Ageing, University of Western Australia, Crawley, WA 6009, Australia; leon.flicker@uwa.edu.au; 5Centre for Aboriginal Medical and Dental Health, University of Western Australia, Crawley, WA 6009, Australia; 6CRANAplus, Cairns, QLD 4870, Australia; dallas@crana.org.au; 7Menzies School of Health Research, Charles Darwin University, Casuarina, NT 0810, Australia; alan.cass@menzies.edu.au; 8School of Public Health, The University of Queensland, Herston, Brisbane, QLD 4006, Australia; 9Medicine, Dentistry and Health Sciences, University of Melbourne, Parkville, VIC 3050, Australia; 10Cairns and Hinterland Hospital and Health Service, Queensland Health, Cairns, QLD 4870, Australia

**Keywords:** health services research, primary health care, dementia, Aboriginal and Torres Strait Islander health, Aboriginal participatory action research, Indigenous quality appraisal, continuous quality improvement, research yarning, health promotion, preventive health

## Abstract

This protocol describes the methodology and methods for a collaborative project with eight Aboriginal and Torres Strait Islander primary health care (PHC) organisations, across three Australian states and one territory, to increase clinical service performance and access to preventive health and health promotion services for preventing, identifying, treating, and managing dementia risk in Aboriginal and Torres Strait Islander communities. Aboriginal participatory action research (APAR) methodology will be the framework for this project, incorporating continuous quality improvement (CQI), informed by research yarning with stakeholder groups, comprising community members and PHC staff and service providers and data collected from the auditing of client health records and the mapping of existing clinical processes and health services at each partnering PHC organisation. The qualitative and quantitative data will be summarised and discussed with stakeholder groups. Priorities will be identified and broken down into tangible PHC organisation deliverable strategies and programs, which will be co-developed with stakeholder groups and implemented cyclically over 24 months using the Plan, Do, Study, Act model of change. Key project outcome measures include increased clinical service performance and availability of preventive health and health promotion services for safeguarding against dementia. Project implementation will be evaluated for quality and transparency from an Indigenous perspective using an appropriate appraisal tool. The project processes, impact, and sustainability will be evaluated using the RE-AIM framework. A dementia safeguarding framework and accompanying tool kit will be developed from this work to support Aboriginal and Torres Strait Islander PHC organisations to identify, implement, and evaluate dementia safeguarding practice and service improvements on a broader scale.

## 1. Introduction

It is a global phenomenon that people are living longer, and the world population is ageing. Australia’s First Nation populations, Aboriginal and Torres Strait Islander peoples, are no exception to this trend, with most recent figures showing significant increases in average life expectancy at birth for both males and females, of 4.4 years and 2.7 years, respectively, in the five-year period prior to 2017 [1]. Despite this, global data show that such achievements are often coupled with increases in morbidity, as well as ageing-related disease and disability, including dementia [2]. Dementia is a neurodegenerative disorder, associated with progressive cognitive and functional impairment. Dementia can occur at any age but is more prevalent in persons aged 65 years and over. In line with global trends, dementia is a leading cause of death in Australia and a major cause of disease burden nationwide [3].

A recent modelling study estimated future growth in the number of older Aboriginal and Torres Strait Islander peoples living with dementia [4]. Projections from this study indicate that by 2051, there will be a substantial growth in the number of older Aboriginal and Torres Strait Islanders peoples. In parallel to this growth is a projected increase in the number of people living with dementia, with this trend evident across all age groups, from 50 years and over. Consistent with First Nation populations around the world, dementia prevalence studies have found this disorder to be already occurring at high rates, and in younger age groups, among Australia’s Aboriginal and Torres Strait Islander peoples [5,6,7]. This highlights an urgent need to address the incidence of dementia, if Australia’s ageing Aboriginal and Torres Strait Islander peoples are to be safeguarded against the impact of this disorder, over the coming decades. 

While age is the strongest known risk factor for cognitive decline, dementia is not an inevitable consequence of ageing. There is an increasing body of evidence supporting a range of potentially modifiable factors that may influence dementia risk. A 2020 systematic review and meta-analysis identified twelve potentially modifiable factors for dementia across the life course that, if addressed, may prevent or delay up to 40% of dementias globally [8]. These include early life education; mid-life hypertension, obesity, hearing loss, traumatic brain injury, and alcohol misuse; and later life smoking, depression, social isolation, physical inactivity, diabetes, and air pollution [8]. The evidence suggests that addressing these factors across the life course may prove protective against the disorder [8]. 

Several cross-sectional studies in the remote Kimberley region of Western Australia, regional and urban areas of NSW, and the Torres Strait Islands have confirmed an association between several of these potentially modifiable risk factors and dementia in Aboriginal and Torres Strait Islander peoples. These include traumatic brain injury [9,10,11], low education [10,11], smoking [11], hearing loss [9,10], high-risk alcohol intake, depression, social isolation, and low levels of physical activity [9]. There is also strong evidence for additional potentially preventable medical risk factors and dementia, including stroke [10,11], epilepsy [9,11], chronic kidney disease, and cerebrovascular disease, and lower-level evidence for the accumulation of vascular factors [10]. A five-year longitudinal study of older Aboriginal and Torres Strait Islander people living in rural and remote areas of the Kimberley, WA, investigated associations between sociodemographic and clinical factors at baseline with cognitive status at follow-up [12]. The longitudinal risk factors associated with a decline from normal cognition to impairment were age and head injury. Other associations with cognitive decline were stroke, head injury, non-aspirin analgesics, lower body mass index, and higher systolic blood pressure. The combination of findings from these studies provides strong evidence linking head injury, stroke, epilepsy, hearing loss, and low education and an increased risk of dementia in Aboriginal and Torres Strait Islander peoples. 

The commonalities in risk factors across these studies are likely reflective of health and social disadvantages experienced by Aboriginal and Torres Strait Islander peoples across the life course, which contribute to the development of chronic conditions in midlife and eventuate into increased dementia risk in later life. However, the evidence from these studies is highly promising in that most of the factors identified are potentially preventable or modifiable through both individually targeted and population-wide interventions and health care. This means a large proportion of dementias in Aboriginal and Torres Strait Islander peoples have the potential to be delayed or prevented through safeguarding against risk factors (e.g., head injury) and adhering to a healthy lifestyle across the life course and the optimisation of primary health clinical care and preventive health and health promotion services.

Addressing risk factors for all chronic diseases, whether at the health systems or population level, is a complex undertaking often intertwined with the broader social and economic environment. Historically, First Nation peoples around the world have been subjected to wide-scale socio-economic disadvantage stemming back from early colonisation. War, displacement, the spread of deadly diseases, forced assimilation, and abolition of social and cultural practices and languages have all significantly impacted the population size of Australia’s First Nation peoples. Racism and discrimination, inadequate housing and health care infrastructure, poor education, low income and employment opportunities, ongoing intergenerational trauma, and a deficit discourse continue to impact the health and well-being of Aboriginal and Torres Strait Islander peoples in Australia today [13,14,15]. In recognition of these complexities, approaches to health promotion and addressing potentially modifiable factors demonstrated to safeguard against dementia should (1) be tailored to communities through the recognition and application of contextual factors, such as the local socioeconomic environment, and (2) utilise a “strengths-based” approach, which focuses on applying the knowledge, attributes, and abilities of the community, to the promotion of health and well-being across the life course on a broad scale. 

In Australia, Aboriginal and Torres Strait Islander primary health care (PHC) organisations offer the first point of contact for coordinated holistic and culturally appropriate clinical care and access to health services for Aboriginal and Torres Strait Islander peoples. These First-Nation-specific PHC organisations are predominantly community controlled but can be government run. All offer prevention and early intervention, as well as the treatment and management of acute and chronic conditions, with an overarching aim of providing non-discriminative and accessible clinical care and health services to promote and strengthen health outcomes for Aboriginal and Torres Strait Islander individuals and populations [16,17]. 

Aboriginal and Torres Strait Islander PHC organisations are, therefore, well placed for delivering strength-based clinical care and health services as part of their broader health strategy to promote positive health and well-being across the life course, to safeguard against dementia in later life. Despite this placement, evidence suggests that due to competing priorities in PHC, providing prevention and early intervention aspects of care can be challenging [17,18]. Furthermore, there is evidence to suggest that limited education and training about dementia for PHC staff may result in it being classified as low priority [19,20].

Participatory action research (PAR) is an approach to public health that is based on iterative cycles of reflection, data collection, and action to address health inequalities [21] and involves the researchers and participants working together to understanding a health issue and identifying and actioning an appropriate solution [21]. PAR seeks to generate knowledge and understanding of the health issue as well as any impacting local contextual factors (e.g., culture, environment). The researcher and participant then work collaboratively together to develop an action for change. PAR involves working together with participants at every phase, from knowledge generation to data collection, interpretation of results, and developing, implementing, and evaluating change. It supports equal and reciprocal relationships between the researcher and participants and builds on the strengths and resources of the community to facilitate the change [22]. Because of this, PAR has been widely promoted as an empowering and effective way for working with Indigenous people to achieve better outcomes in health, education, and community building [23]. However, traditional PAR has recently come under criticism as a “Westernised” approach that fails to take account the distinctive elements that comprise Indigenous knowledge systems [24], those being “epistemology” (Indigenous knowledge), “ontology” (Indigenous ways of being), and “axiology” (Indigenous ways of doing). Aboriginal PAR (APAR), which is recognised as its own distinct research methodology, encompasses each of these knowledge components and applies them in in three distinct ways. First, Indigenous knowledge (understanding constructed through exploration, reflection, and interaction) forms the framework for APAR; second, Indigenous ways of doing (beliefs, behaviours, experiences, and realities) ground APAR; and third, Indigenous ways of doing (values, ethics, protocols, and guidelines) underpin APAR [24]. APAR increases the voice of Indigenous peoples and places it at the centre of research that is about them and has multiple layers of Indigenous governance and reciprocity [24]. Applying APAR methodology decolonises research by enabling Indigenous peoples to take control and ownership of the investigation and self-determine the actions and outcomes. 

Yarning is an Indigenous cultural form of conversation and is the most recommended method for collecting qualitative data with Indigenous groups [25,26]. Yarning enables the sharing of information through storytelling, which may include knowledge, lived experiences, and beliefs [25]. The yarning method is highly suitable for facilitating and supporting APAR due to enabling the recognition and inclusion of Indigenous knowledge systems. The Yarning method is underpinned by formal protocols for culturally safe discussion and can be categorised into four different types. These include “social yarning”, which takes place first and helps to establish connections and respectful communication and build trusting relationships between individuals and the research team; “research yarning”, which is a semi-structured approach to gathering information about the research topic; “collaborative yarning”, which involves a group of people actively engaged in sharing information and discussing ideas to support research output; and “therapeutic yarning”, which follows on from the yarning process and may involve the disclosure of personal or emotional information. The researcher will turn to listening and supporting the participant to voice their story [25]. The combination of these different types of yarning help to produce data that are both rich and authentic. 

Continuous quality improvement (CQI) is a structured organisational process for optimising quality in health care settings. CQI applies a whole of organisation approach to the planning and implementation of health system and process changes to strengthen clinical care, health service delivery, and client health outcomes [27]. It works alongside accreditation, governance, and monitoring, to strengthen quality [28]. Like PAR, data are collected to identify areas of care that may benefit from service strengthening activities, and multidisciplinary teams of staff work together to develop improvement actions, which are implemented and evaluated through continuous cycles of change. The actions implemented as part of this process support PHC organisations to strengthen their capacity to deliver quality, appropriate, and responsive clinical care and health services that align with best practice and meet the needs and priorities of their broader community [24,28,29]. CQI programs have increased in popularity in recent years and are proving to be an appropriate and accepted method for working with Aboriginal and Torres Strait Islander PHC organisations to strengthen clinical care and health service provision on a national scale [30]. 

Plan, Do, Study, Act (PDSA) is a structured four-stage problem-solving model used to support CQI within the health care setting. These stages include “Plan” (an action for improvement is identified and a plan for implementing the change is developed), “Do” (the action is implemented according to the plan), “Study” (the relevant data are collected and evaluated to determine the success of the action), and “Act” (which identifies and implements any necessary modifications required to increase action success) [31]. The PDSA model is used iteratively over a predetermined period as a continuous process for addressing multiple priorities identified for quality improvement. 

This project is led by a national multidisciplinary team of Indigenous and non-Indigenous geriatricians, health practitioners, and research academics from across five states and one territory, some of whom have been delivering clinical services for older Aboriginal and Torres Strait Islander communities for over 25 years and together have an extensive research portfolio on dementia and the health of older Aboriginal and Torres Strait Islander peoples [6,7,10,11,12,32,33]. Aboriginal and Torres Strait Islander research officers and health workers are core members of this team, and each has a lifetime of experience working in primary health care and/or research. It is through the work of this team that communities have identified a need for dementia prevention initiatives to tackle the high rates of dementia found in Aboriginal and Torres Strait Islander communities.

This research protocol describes the methodology and methods for a national project that will be undertaken in partnership with community-controlled and government-run Aboriginal and Torres Strait Islander PHC organisations to strengthen clinical performance and health services for promoting health and preventing, identifying, treating, and managing risk factors for dementia across the life course to help safeguard against this disease in later life. 

## 2. Aims and Objectives

The primary and secondary aims of this project are to work in partnership with Aboriginal and Torres Strait Islander primary health care organisations as follows:

Primary aims:Strengthen clinical performance for preventing, identifying, treating, and managing potentially modifiable risk factors associated with dementia.Strengthen the availability of preventative health and health promotion services for safeguarding against dementia.Integrate clinical performance and service strengthening strategies and programs into ongoing practice and organisation policies and procedures.

Secondary aims:4.Strengthen the capacity of PHC staff to undertake CQI and deliver best practice care for the prevention, identification, treatment, and management of dementia risk in everyday clinical practice.5.Strengthen the capacity of Aboriginal and Torres Strait Islander communities to understand dementia and health promotion approaches to safeguarding against dementia.6.Integrate capacity-strengthening strategies into ongoing organisational practice, policies, and procedures.

The objectives are as follows: Utilise a participatory action research approach to identify and co-develop PHC deliverable dementia safeguarding strategies, programs, and resources.Utilise a continuous quality improvement methodology to implement and evaluate PHC deliverable strategies and programs, for strengthening clinical service performance and the availability of health services for safeguarding against dementia in Aboriginal and Torres Strait Islander communities.Evaluate this approach and methodology for appropriateness, effectiveness, and sustainability in achieving the project aims and objectives and answering the research question.Work with PHC staff to identify opportunities for training on CQI and dementia safeguarding practices.Work with the community to identify opportunities for strengthening the knowledge and understanding of dementia and associated safeguarding strategies.Integrate dementia safeguarding strategies and programs into clinical practice and organisation policies and procedures.Integrate community and staff capacity-building strategies into organisation policies and procedures.Synthesise the evidence into a culturally appropriate framework and accompanying tool kit for Aboriginal and Torres Strait Islander PHC organisations to identify, implement, and evaluate their own dementia safeguarding practice and service improvements.

## 3. Research Question

Is the proposed protocol an appropriate and effective approach for making sustained improvements to clinical performance and the availability of preventive health and health promotion for safeguarding against dementia in Aboriginal and Torres Strait Islander PHC organisations? 

## 4. Methodology and Methods

### 4.1. Design

Aboriginal participatory action research (APAR) and continuous quality improvement (CQI) methodologies, informed by research yarning with stakeholder groups (community members, PHC staff, and service providers), and data collected from the auditing of client clinical records and the mapping of existing clinical processes and health services, at each partnering organisation, will inform the methodological framework for this project. Figure 1. displays an overview of the project design. 

The qualitative and quantitative data collected in phase one and two will be reviewed and discussed with stakeholder groups in workshops at the beginning of phase three. Priorities for action will be identified and broken down with each stakeholder group into tangible outcomes that can be delivered through changes to clinical practice and health service provision. PHC organisation deliverable strategies, programs, and resources will be planned and co-developed with stakeholder groups and implemented cyclically over 24 months using the Plan, Do, Study, Act (PDSA) model for change. Opportunities for training relevant to clinical care and health service strengthening activities will be identified with PHC staff. An evaluation will be undertaken with each organisation in phase four, comprising repeat audits, mapping exercises, and collaborative yarning with stakeholder groups, to determine the effectiveness of CQI activities on achieving the outcomes set by each stakeholder group. Research evidence will be summarised, fed back, discussed, and confirmed with stakeholder groups at each project phase. Indigenous team members will co-facilitate all activities involving Aboriginal and Torres Strait Islander stakeholder groups. An Indigenous reference group (IRG) will be established to provide cultural guidance on key research activities and outputs. 

A process, impact, and outcome evaluation will be undertaken to determine the appropriateness and effectiveness of this research approach for achieving the overall aims and objectives of the project and answering the research question. The Reach, Efficacy, Adoption, Implementation, and Maintenance (RE-AIM) framework will guide this process. The project implementation will also be evaluated for quality and transparency from an Indigenous perspective using the Aboriginal and Torres Strait Islander Quality Assessment Tool [34].

### 4.2. Setting

Eight Aboriginal and Torres Strait Islander PHC organisations in urban and regional QLD, NT, NSW, and WA will be invited to participate in this study. To date, two Aboriginal and Torres Strait Islander PHC organisations, one in urban and one in regional QLD, have provided letters of support and signed agreements to partner with the research team on this project. Approaching potential partner organisations in WA, NT, and NSW will be done pragmatically through pre-existing networks established by the project investigators and Aboriginal and Torres Strait Islander research team members. Peak organisations representing Aboriginal and Torres Strait Islander Community Controlled Health Organisations in each state and territory will be approached to seek their support with this project. Senior project investigators will contact senior management of prospective partner organisations. Face-to-face meetings will be organised with interested organisations to discuss the project in detail. Indigenous team members will accompany non-Indigenous team members on all visits to PHC organisations, lead communication, and attend all online meetings.

### 4.3. Facilitation

A CQI coordinator will be employed part-time with the research funding and based internally at each partnering PHC organisation. The primary role of this individual will be to work with multidisciplinary teams of staff and health service providers to identify, lead, drive, and monitor changes to clinical practice and health service delivery from within their organisation. They will assist with data collection and the interpretation of the findings; co-facilitate yarning sessions and workshops with PHC staff, service providers, and community members; and work closely with these stakeholder groups to plan, co-develop, implement, and evaluate PHC deliverable strategies and programs, as part of the quality improvement process. The CQI coordinator will co-facilitate relevant staff training and information and feedback sessions with PHC organisation staff and service provider and community groups. They will receive necessary training and support from the external CQI coordinator to fulfil activities associated with this role. Ideally, this position will be held by an existing Aboriginal and/or Torres Strait Islander staff member of the partnering PHC, who will have the skills to work collaboratively with multidisciplinary teams of PHC staff and have positive connections with the local community. 

Aboriginal and/or Torres Strait Islander CQI coordinators will be employed by James Cook University as members of the Healthy Ageing Research Team (HART). The responsibilities associated with the external coordinator role are to provide guidance, support, and relevant education and training to the internal coordinators, across all partnering PHC organisations. The external CQI coordinator will assist the internal CQI coordinators with data collection, interpretation of results, strategy development and program co-design, and co-facilitation of yarning sessions and training and education workshops with stakeholder groups, as required. 

### 4.4. Indigenous Reference Group

An Indigenous reference group (IRG) will be formed for consultation on all major research activities, inputs, and outputs associated with this project, such as the suitability of the project methods, the analysis and interpretation of qualitative data, the suitability of existing dementia and related preventive health resources, and the interpretation and write-up of project outcomes. IRG members may also assist with building new relationships between the project team and the PHC organisations and community members. IRG members may include Aboriginal and Torres Strait Islander project investigators and research team members, PHC organisation staff and service providers, community Elders, and dementia advocates. The aim is for the IRG to have local representation of Aboriginal and Torres Strait Islander staff and community members from each partnering organisation. 

The research team will develop a draft “terms of reference” (ToR) document for the IRG to highlight the purpose and goals of the IRG, as well as any anticipated activities, time commitments, and cost reimbursement details associated with meeting attendance. PHC organisations will be asked to email all Aboriginal and Torres Strait Islander staff to notify them of the opportunity. Advertisements will be placed on display in the PHC organisations clinic, social media, and web-based platforms to invite both staff and community members to be involved. Community members attending engagement activities facilitated by the research team, with a keen interest in dementia and preventative health/health promotion, will also be invited to participate on the IRG. All interested parties will be provided with a copy of the IRG ToR for their consideration and will receive follow-up communication from one of the Aboriginal and/or Torres Strait Islander CQI coordinators, who will provide a verbal overview of the ToR document prior to confirmation of interest. 

Representation on the IRG from each partnering organisation and community is essential to ensure project activities are guided by local and contextual factors influencing the organisation and broader community. The IRG will be led by Aboriginal and/or Torres Strait Islander team members. It is anticipated that this group will meet around three times a year from the engagement of IRG members through to the end of the project. The contribution of the IRG will be acknowledged in all project outputs, and IRG members will be offered a gift voucher in appreciation of their time, knowledge, and expertise.

### 4.5. Engagement Activities

The first 12–18 months of this project have been allocated to engagement activities, where the primary focus is on building respectful and reciprocal relationships with Aboriginal and Torres Strait Islander PHC organisations and the broader Aboriginal and Torres Strait Islander community where pre-existing relationships may be limited. Engagement activities may include interactive workshops on healthy ageing and dementia prevention for community and staff and information stalls presenting health information at local and cultural events, as well as “Tea with the Doctor” [geriatrician] events and sharing health messages through local media platforms. Additional events will be guided by each health organisation and community preference. All engagement activities will be co-facilitated and guided by the Aboriginal and/or Torres Strait Islander research team members. Engagement activities will continue across all partnering PHC organisations and communities throughout the project. 

## 5. Aboriginal Participatory Action Research (APAR)

APAR aligns well with each of the NHMRCs eight key principles for the responsible conduct of research (NHMRC 2018) and has been designed specifically as a foundation methodology for research involving Australia’s Aboriginal and/or Torres Strait Islander peoples. Dudgeon et al. 2020 identify seven core components and highlight nine guiding principles of APAR, which need to be considered when applying this methodology to health research with Aboriginal and Torres Strait Islander communities. These include the following: The involvement of Aboriginal co-researchers in supporting communities to collectively identify risk and protective factors;A research process that respects Indigenous peoples as experts-by-experience of their own health, their families, and their communities;Indigenous leadership and governance and the establishment of local Indigenous community reference groups;Localised knowledge generation;Community-level feedback and dissemination;The enactment of the NHMRC Indigenous core values;The application of nine guiding principles: (i) Indigenous health being viewed in a holistic context; (ii) Indigenous choice being a central health service provision; (iii) care being culturally informed; (iv) the impact of colonisation on physical and mental health; (v) human rights being respected; (vi) the impact of racism, stigma, contextual adversity, and disadvantage; (vii) the centrality of Indigenous kinship; (viii) the diversity of Indigenous peoples and their environments and circumstances; and (ix) the recognition of their strengths and abilities and connection to community and country.

APAR methodology will form the framework for this research project, and the seven core components and nine guiding principles will form the foundations of all project methods, activities, and outputs.

## 6. Phase One: Yarning

The yarning method will be utilised with stakeholder groups to explore opportunities, approaches, and challenges to safeguarding communities against dementia. 

Participants: Those meeting the eligibility criteria of the following stakeholder groups will be invited to participate:Aboriginal and/or Torres Strait Islander Elders, community members, and dementia advocates, aged 18 years and over, who are a regular client at the participating PHC organisation;Aboriginal and/or Torres Strait Islander PHC staff including management, administration, clinical and non-clinical staff, and service providers, with the aim to have mixed representation from each group.

Yarning sessions will be held separately for the different stakeholder and demographic groups (e.g., age, gender), as required. A minimum of one yarning session will be held with each stakeholder group, at each setting. The yarning sessions will be capped at 10 persons per session to ensure all voices are able to be heard. Additional yarning sessions will be held where there is interest. 

Recruitment: Opportunistic and purposive sampling methods will be applied to the invitation of Aboriginal and/or Torres Strait Islander community members to participate in the yarning sessions. A purposeful sampling technique based on the desired sample characteristics will be used to invite community participants [35]. A mix of male and female, Elder, and general community members across different age groups will be approached to participate in the yarning sessions with the community. The CQI coordinator will consult with the PHC organisation manager and staff to identify regular clients who are attending the clinic and meet the eligibility and target criteria to approach for participation. Advertisements will also be placed in the clinic, on relevant social media platforms, and on local community information boards. 

The purposeful sampling method will also be applied to the invitation of the staff stakeholder group. Aboriginal and/or Torres Strait Islander PHC staff and service providers, with a mix of males and females from the different multidisciplinary teams (e.g., practice managers and administration staff, general practitioners and nurses, health workers, and allied health), will be invited to participate in the yarning sessions by the CQI coordinator. However, as the CQI methodology relies on a “whole of team approach” to implement effective and sustainable change to PHC settings, and the data collected during the yarning session will be used to guide priority setting, it is essential that all staff, regardless of their Indigenous status, are included in this process. The PHC organisation administration team will be asked to email out the invite to all staff and relevant service providers to invite them to participate, and advertisements will be placed in staff communal areas. 

Facilitation: The yarning sessions will be co-facilitated by the internal and external Aboriginal and/or Torres Strait Islander CQI coordinators and any interested PHC staff members. Training on research yarning and facilitation methods will be provided to CQI coordinators, research staff, and PHC staff, as required, and delivered by one of the Aboriginal team members with experience applying this method in research. The facilitators will be provided with a verbal overview and written outline of the proposed yarning process and topics prior to the session. 

Consent: Individual informed consent will be sought from the community participants. The facilitators will provide a verbal plain language statement of consent to participants before commencing the research yarn. They will address any questions arising and support participants to understand and complete consent, as required.

Process: Yarning will commence with a social yarn over refreshments to establish connections and respectful communication and build trust between the research team members and community. Local co-facilitators will commence the more formal yarning with “welcome to country” followed by an introduction to the project. Co-facilitators will then introduce clinical members of the research team, who will provide an overview of dementia and its associated protective and risk factors in layperson language and answer participant questions about dementia. Facilitators will then commence the research yarn to explore the research topics through broad opening questions to encourage participant discussion and stories. This will be followed by a collaborative yarn, which will involve the facilitators working with the stakeholder groups to identify up to five priorities for action and discuss ideas for health service deliverable strategies and/or programs to address the priorities (e.g., health promotion program, education, and increased annual health checks). The session will end with a therapeutic yarn, offering participants the opportunity to discuss more personal or emotional issues with the facilitators and/or clinical members of the team that may have arisen during the session. 

Analysis: Qualitative data from the yarning circles will be transcribed and coded. Reflective thematic analysis [36] will be used to identify themes, patterns, and relationships in the data. NVivo 12 qualitative data software V12 (QSR International) will be used to manage the data. 

The analysis of qualitative data will be three-phased: first, a reflective thematic analysis will be applied to explore the contextual factors impacting the health and well-being of the local Aboriginal and/or Torres Strait Islander communities (the nine guiding principles for APAR will be used as the framework for this analysis); second, the data will be explored to identify opportunities, approaches, and challenges to dementia safeguarding practices in the local Aboriginal and Torres Strait Islander communities; and third, these two sets of data will be analysed to explore the relationships between dementia safeguarding practices and local contextual factors that may support or inhibit clinical care and engagement with health services. The Aboriginal and/or Torres Strait Islander research team members and CQI coordinators will guide and support the analysis and interpretation of qualitative data. The IRG may also assist with this process. 

This evidence will be summarised for inclusion in the written summary report that will be used to support PHC organisation priority selection and outcome setting in phase three and will be applied to the co-development of strategies and programs to support CQI activities at the PHC organisation that are appropriate and relevant to the local community. 

## 7. Phase Two: Quantitative Data Collection

### 7.1. Clinical Records Audit

The auditing of clinical records is an essential component of the CQI process [37,38]. Audit data will be collected at baseline to provide evidence on the current clinical service performance measured against best practice and client health outcomes measured against health indicators. These data will be used to work with PHC organisation staff to identify strengths, gaps, and limitations in the current clinical practice and health service provision. 

A standardised audit tool has been developed based on evidence derived from a scoping review of existing audit tools for quality improvement, evidence-based clinical guidelines, and quality indicators for Aboriginal and Torres Strait Islander PHC. The audit tool will collect information on clinical performance and health indicators from nine clinical and care domains (see Table 1). All items in the audit tool relate to protective and/or risk factors associated with dementia in either the global population and/or Aboriginal and Torres Strait Islander populations and align with the clinical best practice guidelines. The tool will be tested and evaluated for validity prior to use. 

The audit tool will be situated in the REDCap electronic data capture tool hosted at James Cook University. REDCap (Research Electronic Data Capture, https://www.project-redcap.org/, accessed on 15 October 2023) is a secure, web-based software platform designed to support data capture for research studies, providing (1) an intuitive interface for validated data capture, (2) audit trails for tracking data manipulation and export procedures, (3) automated export procedures for seamless data downloads to common statistical packages, and (4) procedures for data integration and interoperability with external sources [39,40]. The audit tool is accompanied by a reference guide, which includes instructions for data collection and a cross-reference of the evidence base. 

Auditors: For reasons relating to client privacy and confidentiality, people accessing the PHC clinical management software to collect data for the audits must be current PHC organisation staff members. It is anticipated that the internal CQI coordinator, with the support of another staff member, will be responsible for collecting data from their organisation’s clinical records. 

Audit training and data consistency checks: The auditors will each receive training on the audit tool and the relevant clinical record management system, as required. The audit tool guide will be used to facilitate training and for cross-checking throughout the audit process. The first ten health record audits at each PHC organisation will be completed solo by the CQI coordinator. A second staff member will complete the same ten audits. The data from the duplicate audits will be cross-checked for inconsistencies. Further training and/or guidance will be provided to address any data inconsistencies. The audit training, guide, and duplicate cases are to strengthen the quality of the data being collected. 

Participants: Between 115–155 audits will be undertaken at each PHC organisation using the standardised audit tool. The organisations’ electronic clinical records software will be used to select clients at random based on the following criteria: 

Eligibility criteria: Clients must be Aboriginal and/or Torres Strait Islander, aged 18 years or over, and be classed as a regular client (≥3 visits over the past 24 months). A similar distribution of males/females and age groups (18–44; 45–59 and 60+) will be sought. 

Sample size calculation: The number of clinical records to be audited at each PHC organisation will be calculated based on the total number of regular clients (3+ visits within the previous 24 months), aged 18 years or over, at each setting, and a representative figure for pre-intervention screening and assessment rates for mild cognitive impairment (MCI) and dementia in Aboriginal and Torres Strait Islander PHC organisations. 

The “Let’s Chat Dementia” project collected baseline data from audits of the clinical records of Aboriginal and Torres Strait Islander PHC clients, aged 50 years and over, from 11 Aboriginal and Torres Strait Islander PHC organisations from across four Australian states/territories. The preliminary figures (unpublished) from the baseline data suggest that for every 100 clients, 13 clients had been screened or assessed for MCI or dementia using a standardised tool within the previous 12 months. That results in an average screening rate of 13% across all Aboriginal and Torres Strait Islander PHC organisations. The preliminary figures from the 4-year follow-up data suggest screening and assessment rates for MCI or dementia using a standardised tool increased by 92%, from 13% to 25%.

Based on these figures, the following calculation will be performed at each PHC organisation to determine the total number of audits to be completed. The confidence level will be set at 95%, the population (N) will be the current number of clients at the PHC organisation meeting the audit eligibility criteria, the screening rate will be set at 13%, and the confidence interval (CI) will be set at 0.05. It is estimated that a sample size of between 115 and 155 records will be audited at each partnering PHC organisation, which would allow for the detection of a significant increase at *p* = 0.05 in the proportion of clients being screened for MCI/dementia at each setting. We may also expect to see around a 90% increase in the number of clients aged 45+ being screened for MCI/dementia with a standardised tool at each setting. 

### 7.2. PHC Organisation Survey

The PHC organisation survey will be completed at baseline, with each partnering PHC organisation. The survey comprises a brief set of questions about each organisation’s current operating environment and collects information on their location, service population, and governance arrangements. The PHC organisation survey has been adopted from an existing suite of tools to support quality improvement in Aboriginal and Torres Strait Islander primary health care settings [41]. 

### 7.3. Health Service Mapping

The mapping of organisation systems and the process for providing clinical care and services relevant to safeguarding against dementia will be undertaken with each PHC organisation. A catalogue of internal and external services (e.g., diabetes management programs, smoking cessation, hearing screen, etc.) accessible through the health organisation and/or community will be compiled. 

### 7.4. Data Analysis

Quantitative data from the clinical records audit, PHC organisation survey, and health service mapping will be entered into SPSS Statistics for Windows (SPSS Inc., https://www.ibm.com/products/spss-statistics, accessed on 15 October 2023) for analysis. Summary and inferential statistics will be used to describe and analyse the data. Audit data will be stratified by demographics and the absence or presence of risk factors and/or diseases. Actions by the health organisation in response to an identified risk and/or disease, including intervention, treatment, management, and monitoring, will be assessed against practice guidelines for Aboriginal and Torres Strait Islander primary health care [42,43]. 

These results, along with the qualitative findings and feedback on stakeholder priorities and proposed outcomes from phase one, will be compiled into a written summary report for the reconfirmation of priorities, outcomes, and action planning with staff stakeholder groups in phase three. 

## 8. Phase Three: CQI Activities

Evidence from the written summary report will be presented back to each stakeholder group for review and discussion. The CQI coordinators will work with these groups to confirm up to five priorities each for action and identify tangible PHC organisation deliverable solutions (e.g., health promotion strategies) and outcome measures (e.g., increase annual health check numbers by 50%). 

### PDSA Cycles

The PDSA cycles of change are the proposed method to support CQI within each partnering PHC organisation. While PDSA is the suggested model for this project, alternative models better suited to an organisation may be identified and applied. The length of each PDSA cycle will be determined by the complexity of the chosen action to be implemented. This may include short rapid cycles from one to six months’ duration for easy-to-implement changes (e.g., review and modify the current strategy for identifying smokers and referrals to the quit smoking program and implement changes) or longer cycles of six to twelve months’ duration for more complex changes (e.g., co-develop with stakeholder groups a multifaceted cognitive reserve promotion program for older adults and implement the program within the organisation). Cycles may incorporate the implementation of a single broad-scale strategy or program or multiple small-scale strategies. It is proposed that PDSA cycles will occur over 24 months (two years) within each partnering PHC organisation. 

Step 1: Plan 

A multidisciplinary planning workshop will be held with PHC organisation stakeholder groups including senior management, clinical and non-clinical staff, and service providers, ideally, with representation from each discipline operating within the organisation. The workshops will be co-facilitated by the CQI coordinators. The aim of the workshop is for the coordinators and workshop participants to

(i)Review and discuss the qualitative and quantitative evidence in the baseline findings summary report and work together to identify the strengths, limitations, and gaps in current clinical performance and health services;(ii)In consideration of all the evidence, identify opportunities for clinical and/or health service strengthening activities, confirm one to two priorities for action, and break them down into tangible outcomes;(iii)Work together as a multidisciplinary team to develop an action plan to address each priority, including the aim, objectives, and outcome measures, an appropriate strategy or program to implement, the resources and steps required for implementation, a subset of data (e.g., smoking status and health organisation response) from the clinical records to be collected and reviewed to measure progress, and opportunities to merge with existing policies and processes.

Step 2: Do

Each PHC organisation will implement their chosen action for change according to the plan (step 1). The CQI coordinators will work with PHC organisation stakeholder groups to identify and organise opportunities for training and offer continued guidance and support with the change process. 

Step 3: Study 

The chosen sub-set of data to measure progress and outcomes will be collected and analysed. The CQI coordinators will review the data to identify how the actioned change progressed; whether the predefined aims, objectives, and outcomes were met; and if any modifications are required. Enablers and barriers to implementation will be explored through collaborative yarning with staff and community stakeholder groups. The acceptability and suitability of the strategy or program and its content by stakeholder groups for achieving the desired outcome will also be explored. 

Information about enablers will be used to support additional cycles and solutions to barriers discussed and implemented. 

Step 4: Act

Any additional modifications based on the outcome of the previous step will be implemented before the next PDSA cycle of change is commenced, then step one will be reverted to. 


Education workshops


Interactive workshops with the theme of dementia, and health promotion across the life course to safeguard against dementia, will be offered to communities throughout each phase of this project. The priorities identified by the community in phase one will be utilised to guide the content of workshops from phase three onward. The education workshops will be co-facilitated by the Aboriginal and/or Torres Strait Islander CQI coordinators and members of the research team. 


Co-design


When utilised to optimise health care and/or health services, co-design refers to the application of consumer research in the development of an action to promote a desired health outcome [44]. This approach enables consumers to become partners in the improvement process [45]. Involving community and PHC staff and service providers in this process will help to ensure actioned activities and research outputs are tailored to both the PHC organisation and the broader community. 

CQI coordinators will work closely with sub-groups of staff, service providers, and/or community members to review and/or modify existing or develop new dementia safeguarding strategies, programs, and resources for implementation through the PHC organisation. Existing co-design tools and resources [45] will be utilised to support this process. Contextual qualitative evidence from phase one will also be applied to guide the co-design of project outputs. Community members attending engagement activities and yarning sessions expressing a keen interest in dementia and/or preventative health/health promotion will be invited to participate in co-design workshops. Multi-disciplinary teams of staff and service providers working in the area relevant to the strategy or program being co-designed will be encouraged to participate in co-design workshops. 

## 9. Phase Four: Evaluation

The collection of subsets of data at the end of each PDSA cycle will allow the tracking of progress over time and determine the effectiveness of the activity on outcome measures at each partnering organisation. The data subsets will relate directly to the change being implemented (e.g., the number/proportion of annual health checks completed). A written log and feedback from stakeholder groups will be used to identify facilitators and barriers to implementation and sustainment of the change within each partnering organisation. 

## 10. Project Evaluation

An evaluation of the overall project processes and impact will be undertaken based on the Reach, Efficacy, Adoption, Implementation, and Maintenance (RE-AIM) framework [46]. This framework is designed to enhance the quality, speed, and public health impacts of efforts to translate research into practice in the following five steps: (1) reach of the intended target population, comprising engaging and building the capacity of PHC staff to undertake CQI and client receipt of dementia safeguarding clinical care and health services; (2) efficacy of the intervention, comprising the effectiveness of CQI activities on clinical performance and access to health services; (3) adoption of the intervention, comprising health system integration of the strategy or program and its application by PHC organisation staff and service providers; (4) implementation, comprising consistency and modifications of the project protocol made during delivery; and (5) sustainment, comprising intervention effects on clinical performance and availability of health services over time [46] and integration into existing practice and policies. Table 2 outlines the evaluation measures and data sources for each evaluation parameter for this project. 

The project evaluation will involve a mixed methods approach including repeating the baseline data collection at 24 months, comprising a clinical records audit, a PHC organisation survey, health service mapping, and yarning sessions with each stakeholder group. Quantitative and qualitative data will be analysed for changes to clinical service provision and the availability of health services from the baseline to 24 months to evaluate the overall effectiveness of the project for achieving the desired project aims and objectives, answering the research question, and achieving the short-term and mid-term project outcomes. 

### Project Outcomes


Short-term outcomes


To strengthen: IWorkforce capacity of the partnering Aboriginal and Torres Strait Islander PHC organisation to provide clinical care and health services in accordance with clinical best practice guidelines [42,43] for the prevention, early identification, treatment, and management of risk factors associated with dementia in Aboriginal and Torres Strait Islander peoples;IIThe knowledge and capacity of Aboriginal and/or Torres Strait Islander peoples to understand dementia, recognise protective and risk factors, and respond with appropriate preventative health and health promotion strategies to help safeguard themselves, their kin, and the broader community against dementia in later life.


Mid-term outcomes


To strengthen:IClinical service performance for the prevention, early identification, treatment, and management of risk factors associated with dementia in Aboriginal and Torres Strait Islander peoples;IIThe availability of health services for promoting health and preventing or managing risk factors associated with dementia in Aboriginal and Torres Strait Islander peoples;IIIIntegration of changes to clinical service performance and available health services for safeguarding Aboriginal and/or Torres Strait Islander clients against dementia in later life.

As the CQI activities undertaken for this project will be predominantly guided by stakeholder groups at each setting, additional outcomes will be developed with each stakeholder group that reflect the priorities set.

The schedule of proposed tasks and time frames for working with each partnering PHC organisation is outlined in Figure 2.

## 11. Aboriginal and Torres Strait Islander Research Quality Evaluation

The project implementation will also be evaluated for quality and transparency from an Indigenous perspective using the Aboriginal and Torres Strait Islander Quality Assessment Tool [34]. The tool is composed of 14 question domains that assess the quality of health research from an Aboriginal and Torres Strait Islander perspective. The domains include setting appropriate research questions, community engagement and consultation, research leadership and governance, community protocols, intellectual and cultural property rights, the collection and management of research material, Indigenous research paradigms, a strength-based approach to research, the translation of findings into policy and practice, the benefits to the participants and communities involved, and capacity strengthening and two-way learning. 

Applying this method to the evaluation will enable an assessment of the proposed research framework from an Aboriginal perspective to determine whether the project implementation and outcomes align with and reflect Indigenous knowledge systems, values, and principles for ethical research.

It is through each of the proposed evaluation components that the research team will determine whether the proposed protocol is an appropriate and effective framework for implementing effective and sustainable changes to strengthen the clinical performance and the availability of health services for safeguarding against dementia in Aboriginal and Torres Strait Islander PHC organisations.

## 12. Data Management

A data management plan outlining the procedure for collecting, interpreting, and analysing, reviewing, revisiting, and finalising research findings, authorship and attribution, Indigenous data governance, storing and archiving of data, and data access and use will be developed for the project, in consultation with Aboriginal and Torres Strait Islander team members, the PHC organisations, and the IRG. This plan will align with “The guide to applying the AIATSIS Code of Ethics for Aboriginal and Torres Strait Islander research” [47] and the NHMRC “Management of Data and Information in Research: A guide supporting the Australian Code for the Responsible Conduct of Research” [48].

## 13. Project Outputs

There are several expected outputs for this project, which include a culturally appropriate framework and accompanying tool kit for Aboriginal and Torres Strait Islander PHC organisations to identify, implement, and evaluate their own dementia safeguarding practices and service improvements. The toolkit will include a published audit tool and guide; CQI resources and training materials; culturally appropriate dementia-related resources, training, and education materials; and PHC deliverable co-developed strategies and programs for safeguarding against dementia, as well as a scoping review of Indigenous-specific PHC deliverable dementia safeguarding programs and resources (in progress).

Other outputs will include a summary report for stakeholders, highlighting the key findings of the overall project in lay language terms, and several peer-reviewed publications.

## 14. Dissemination

The overall project findings will be disseminated to the participating PHC organisations in a variety of formats, including presentations and a summary of findings report in written and infographics formats. The final framework and accompanying toolkit for Aboriginal and Torres Strait Islander PHC organisations to strengthen clinical performance and availability of health services to help safeguard against dementia in Aboriginal and Torres Strait Islander communities will be made accessible online through key health information websites, National Aboriginal Community Controlled Health Organisations, dementia advocacy organisations, and other relevant agencies. All stakeholders will receive a final project report.

Findings and outputs will be disseminated broadly in a variety of different formats such as presentations at informal community sessions, newsletters, and local media, including social media. Infographic posters summarizing the key messages will be distributed to PHC organisations for display within their service and broader community. Project team members will facilitate the dissemination of the findings nationally and internationally through their existing networks and through conference presentations and peer-reviewed journals.

## 15. Limitations

When planning a research project involving Aboriginal and Torres Strait Islander peoples, it is best practice to scope out research partners at the time of project conceptualisation. This allows for their knowledge and resource requirements to be incorporated and accounted for in the planning and development and research funding application phases.

While a PHC organisation from urban Queensland has been involved from conceptualisation, a limitation is that PHC organisations from remote locations or other proposed states and territory were not engaged in the planning and development phase. To compensate for this limitation, the actions taken are four-fold: (1) the methodology and methods selected for this project are Indigenous research methods; (2) Aboriginal and Torres Strait Islander PHC staff, service providers, and community members from each location will be actively engaged throughout each phase of the project; (3) an IRG will be set up with representation from each PHC organisation and their broader community; and (4) the research protocol will be flexible to accommodate for any contextual factors that may influence an organisation or community’s ability to follow this protocol.

## 16. Conclusions

It is anticipated that the proposed outputs from this project will be useful resources for Aboriginal and Torres Strait Islander PHC organisations on a national level to strengthen clinical performance and health services for promoting protective and preventing and managing risk factors for dementia to help safeguard Aboriginal and Torres Strait Islander communities against increasing rates of dementia over years to come. This project aligns with the broader work conducted around the prevention, identification, treatment, and management of dementia and inter-related chronic kidney, vascular, and metabolic disease impacting First Nation Australians [49,50,51,52,53,54]. It complements the Australian Government’s commitment, outlined in the National Agreement on Closing the Gap, to work in close collaboration with Aboriginal and/or Torres Strait Islander people, communities, and organisations to overcome the inequality they experience and achieve life outcomes equal to all Australians [55]. Furthermore, in alignment with the National Aboriginal and Torres Strait Islander Health Plan 2021–2031 [56], this research focuses on preventing health issues before they occur and optimising the capacity of the Aboriginal and/or Torres Strait Islander health work force to deliver best practice health care. Ultimately, this research project will strive to deliver project outputs and outcomes that are appropriate, beneficial, and safe for partnering Aboriginal and Torres Strait Islander PHC organisations and the individuals and communities they represent.

## Figures and Tables

**Figure 1 mps-06-00103-f001:**
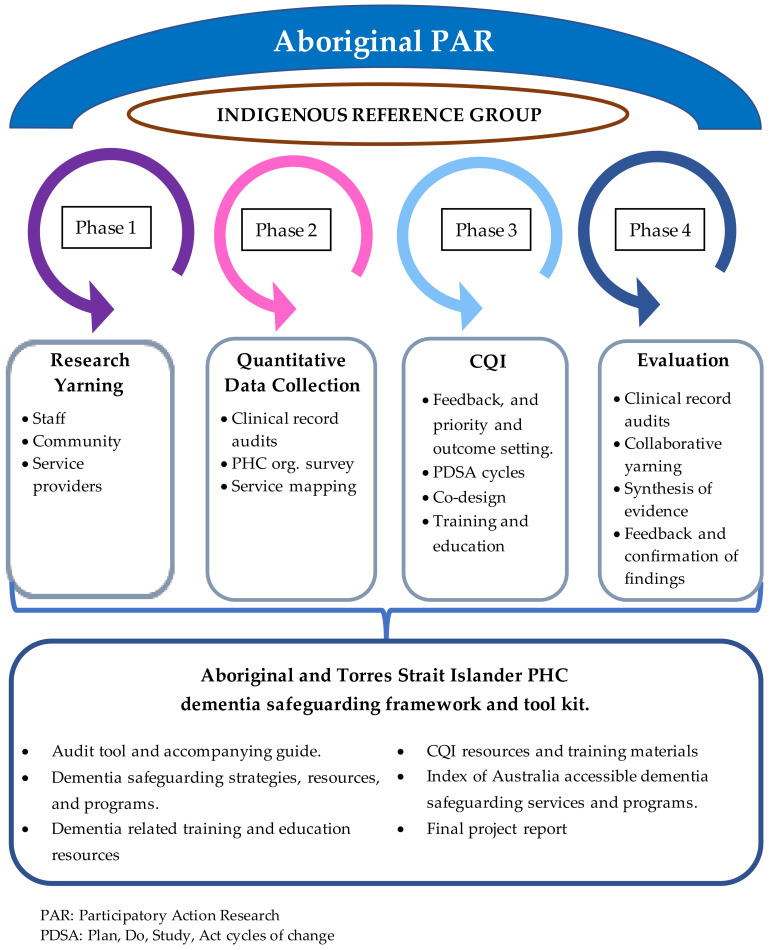
Overview of the project design.

**Figure 2 mps-06-00103-f002:**
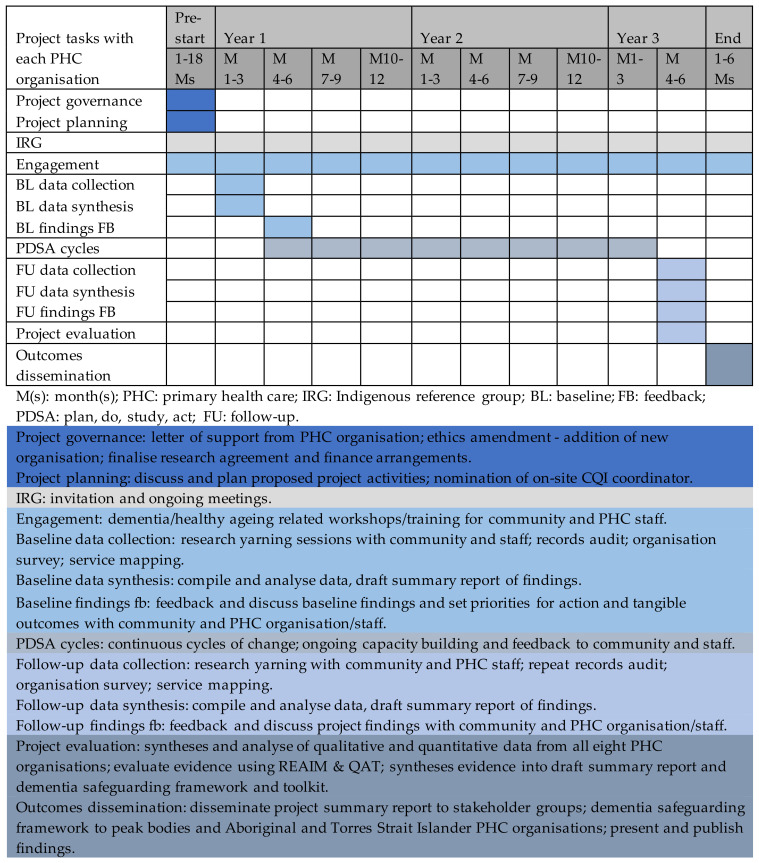
The schedule of proposed tasks and time frames for working with each partnering PHC organisation.

**Table 1 mps-06-00103-t001:** Clinical and care domains of the dementia safeguarding audit tool.

1. Basic demographic information. 2. Health service utilisation. 3. Conditions, diagnoses, and family health history. 4. Medication (current) polypharmacy and medication review. 5. Clinical measurements, investigations, intervention, treatment, management, and monitoring.	6. Hearing and vision screening, intervention, and monitoring. 7. Mental health, emotional and social well-being screening, intervention, treatment, management, and monitoring. 8. Cognitive function screening, assessment, treatment, management, and monitoring. 9. Support services assessment and utilisation.

**Table 2 mps-06-00103-t002:** RE-AIM evaluation parameters.

Evaluation Parameters	Evaluation Measures	Evaluation Data Source
**Reach**	The number, proportion, and representation of PHC organisations partnering on the project; PHC organisation staff and service providers participating in CQI activities, training, and education workshops; Community members participating in CQI activities and attending education workshops and feedback sessions; Clients in receipt of dementia safeguarding clinical care and health services (a target may be set to measure the intended versus the actual reach of dementia risk reduction strategies).	PHC organisation and population demographic information collected by the PHC organisation survey; Records of staff and service providers participating in yarning sessions, priority setting and planning workshops, and IRG and other working groups; Records of staff participating in training and education sessions; Record of community members participating in yarning sessions and priority setting, co-design workshops, IRG, and education and feedback sessions; Clinical records audit at baseline and 24 months.
**Effectiveness**	Impact of CQI activities toward achieving project outcomes: Strengthen clinical service performance for the prevention, early identification, treatment, and management of risk factors associated with dementia in line with best practice guidelines Strengthen the availability of preventive health/health promotion services for safeguarding against dementia.	Clinical records audit at baseline, 24 months, and subset audits to monitor changes in health service performance and health indicators; The addition of health services, measured through health service mapping at baseline and 24 months; Yarning sessions and questionnaires with stakeholder groups to evaluate the effectiveness and suitability of the strategies and programs and their content for achieving the desired outcomes.
**Adoption**	The number and representation of PHC organisation staff and service providers delivering dementia safeguarding service improvements; Consistency and continuity in the delivery of service improvements in line with best practice guidelines; The number and representation of clients accessing best practice clinical care and dementia safeguarding preventive health or health promotion services.	Chart audits at baseline and 24 months; Collection of subsets of audit data.
**Implementation**	Project activities and outputs delivered as intended; Modifications made to the delivery of the project parameters (project input, activities, outputs, and outcomes as described in the project protocol); Identification of barriers and enablers to service improvements; Identification of apparent differences between PHC organisations regarding project parameters and how they are delivered; Aboriginal and Torres Strait Islander research quality evaluation including community engagement and consultation, research leadership and governance, community protocols, intellectual and cultural property rights, the collection and management of research material, Indigenous research paradigms, a strength-based approach to research, the translation of findings into policy and practice, benefits to participants and communities involved, and capacity strengthening and two-way learning.	Comparison of the intended against the actual project parameters and a description of any modifications made and why; Verbal and written feedback from stakeholder groups on barriers and enablers to implementation and the acceptability and suitability of the strategy or program being implemented, as well as a description of the methods and strategies that worked well and not so well and a description of any modifications made; Comparison of project parameters between clusters and sites including a description of what the differences were, why they were made, and how they were addressed; Aboriginal and Torres Strait Islander quality assessment tool against all research evidence relating to how the project protocol and each of the project activities were implemented.
**Maintenance**	Integration of service improvements into clinical practice and health service delivery; Integration of service improvements into organisation policies and procedures; Sustainability of all service improvements.	Documentation of changes to policies and procedures relating to service improvements; Chart audits and health service mapping at baseline and 24 months to determine the effectiveness of the service improvement since the policy or process change; Integration and sustainability of service improvements will also be explored qualitatively through yarning with stakeholder groups; Sustained application of service improvements measured by chart audit and health service mapping at 24 months. Follow-up data will be compared with baseline and subsets of data.

## Data Availability

The data are not publicly available due to Indigenous data sovereignty.

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
