# Peer review of "Safeguarding against Dementia in Aboriginal and Torres Strait Islander Communities through the Optimisation of Primary Health Care: A Project Protocol"

_mps, 2023, doi:10.3390/mps6050103_

Round 1

Reviewer 1 Report

Safeguarding against dementia in Aboriginal and Torres Strait Islander communities through the optimization of primary health care: a project protocol

 Yvonne Hornby-Turner, Sarah G Russell, Rachel Quigley, Veronica Matthews (Quandamooka), Sarah Larkins, Noel Hayman, Prabha Lakhan, Leon Flicker, Kate Smith, Dallas McKeown, Diane Cadet-James, Alan Cass, Gail Garvey (Kamilaroi), Dina LoGiudice, Gavin Miller and Edward Strivens

Jorunal: MPs (ISSN 2409-9279)

Manuscript ID: mps-2574182

Reviewer general comment

This manuscript is an original contribution to the field of Indigenous Health. This protocol describes the methods for a collaborative national project with Australian Aboriginal and Torres Strait Islander Primary Health Care organizations. In the project, Yvonne Hornby-Turner et al., aim to implement and evaluate strategies and programs for optimizing primary health care that promotes prevention of risk factors associated with dementia in these communities.  For this, they will develop a dementia safeguarding framework and toolkit. They will use a Participatory Action Research design, Continuous Quality Improvement methodology, among others, and data will be collected from the auditing of client health records and health services.

In particular, I found this project as highly valuable as it will show multidisciplinary collaboration of Indigenous and Non-Indigenous health workers and researchers. The article is well-written and it will provide of community level data on the availability and effectiveness of health services and promote dementia prevention initiatives to help to overcome the in-equality they experience. I found the field of study to be significant and of interest for readers of Methods and Protocols. I have only a few comments and suggestions listed below, in no particular order.

Comments and suggestions:

1-      The abstract could be more concise.

2-      There seems to be a numbering error in the subtitles starting from section 3 and onward. Please review and correct.

3-      Regarding section 3.2 (Setting): It would be beneficial to provide detailed explanations and possibly create a monthly chronogram outlining proposed activities to ensure that others can replicate the protocol, if necessary.

4-      Overall, I find this article to be very well-written, and I recommend it for publication.

Reviewer 2 Report

I thank and congratulate the authors and the whole research team and beyond that have been involved in writing this protocol and designing the study. As presented, this is a strong, highly relevant, scientifically-informed, protocol. It speaks of an extremely important issue, relevant to the targeted communities/partners, and the protocol, has written, also show a strong, established, relationship between indigenous and non-indigenous researchers, providers, and community members.

I do, however, have some questions, mostly of clarification, about how the protocol came about, as well as a few suggestions on how specific aspects of the current protocol could be clarified, for readers.

But my first question, which can likely be answered by adding clarification on these aspects in the protocol, pertains to why it is the fact that this protocol, methodology and methods selected, remains highly 'non-Indigenous': largely relying on Non-Indigenous research methods and methodology, how 'data' is collected, conceived; analysed (both quantitative and qualitative). For example, I am surprising in reading this protocol that the qualitative methods are not more developed. The yarning method, for example, relies solely on discursive data, that it plans to transcribe. The proposed analyses are also quite traditional (for non-indigenous, positivist research). I would like to hear more from the authors how why these are the methods and methodology that were retained, rather than more Indigenous based approaches, methods, as well as epistemology.

Perhaps, too, say some more about the existing relationships: some are mentioned, as well as the fact that there are some Indigenous research members and collaborators - but can you say a little more about the existing relationships with the larger communities? 

Then, overall, apart from my curiosity about the rationale between this proposed protocol, methods and methodology, I had a few more questions/suggestions to help improve the protocol:

Hypothesis - I find the hypothesis, as written, imprecise. To say the approach used will lead to 'increase PHC' is not clear to me. It can mean a number of things. It would be helpful to be much more specific here. More than one, of course, can be formulated. These could also be formulated with the communities, e.g. during the yarning sessions.

Setting (3.2) the protocol mentions about 12 to 18 months to create new relationships with new communities, where the project could be done. This doesn't seem like much time. Perhaps there are some already exisiting links/relationships? It would be good to mention; otherwise, I don't think this is sufficient time; I would also say the protocol needs to mention that concrete actions, steps and activities will be done during these months to approach the communities, build the relationships. Furthermore, given what we know about the importance of 'doing with' when it comes to (health) research with aboriginal communities, how does the team anticipate revising this protocol following these contacts? For example, what (if any) would be the role of the 'Indigenous Reference group' with regard to approaching new sites?

"Yarning method": could you precise how many people will participate, per group? Also - why only two groups? this doesn't seem enough/sufficient? How long will the yarns last (each session)? Where will they be hosted?

Also -- what do you see are the main differences between the 'yarning method' and 'open or semi-structured qualitative interviews' (group interviews or focus groups). Will there be any visual/art-based approach integrated in the yarn? I was surprised by the limitedness of this phase: only 2 groups, one session; only text-based transcription of the exchanges. I was wondering why these do not occupy a stronger place in the protocol; and why they do not draw more strongly from the yarning methodology (including more than only transcribe the discussions).

 I look forward to seeing a revised copy of the protocol. thank you!

Reviewer 3 Report

I would like to thank the editors and authors for the opportunity to review the article “Safeguarding against dementia in Aboriginal and Torres Strait Islander communities through the optimisation of primary health care: a project protocol”

In general terms, I can say that the article presents specific references of interest, and current ones, 59% are less than five years old.

The theme chosen is interesting and of greater importance, knowing that prevention and protection against dementia are crucial aspects of primary health care, especially because many risk factors for dementia can be managed or reduced with appropriate interventions.

I will now offer some contributions or suggestions for improving the manuscript:

Due to a lapse, the numbering in point 3 is not correct, it goes from point 3.2 to points 4.3 and 4.4. In this sense, it is necessary to review the numbering.

It is suggested that authors describe the estimated number of participants needed to achieve study objectives and how it was determined, that is, how it will calculate the sample size.

The aurors state that "The aim is for the IRG to have representation of staff and community members from each participating PHC organisation and their as sociated community." (line 266-267) In this sense, it seems to me that it would be important to describe how they plan to achieve this representation of both employees and community members and how they will calculate it.

In line 392 the authors state that "Participants: 150 audits will be undertaken at each participating PHC organization using the standardized audit tool.". Clarify how they arrived at the number of 150 audits.

Clarify plans for data entry, coding, security, and storage.

Final decision:

The manuscript needs minor changes.

Thank you very much.

Best Regards

Reviewer 4 Report

Dear authors, congratulations on the subject of the project protocol of the manuscript, as it is of importance about implementing Safeguarding measures against dementia, combining the Continuous Quality Improvement (CQI) methodologies, PDSA cycle and Participatory Action Research (PAR) as a mixed approach (theoretical and practical).

After reviewing the manuscript, I submit the following comments.

Best regards,

In the Introduction section

The bibliography used is generally up-to-date and from different disciplines (for example, social work). However, it is noted that the data on health status (for example, 5th reference) is not updated. I would advise you to find more up-to-date information.

In the Method section

In the “4.5.3. Phase three: CQI activities” subsection

In lines 478 to 479, reference is made to a co-design of the educational workshops between their organizers, with the aborigines. This raises a question for me: would the aborigines who collaborated have a minimum of knowledge about workshop topics such as the health agents, non-professionals with basic knowledge in health care, or would the co-design be open to any aborigine? I recommend that you clarify it within the manuscript to have more exact information about the project.

In the “4.5.4. Phase four: Evaluation” subsection

In lines 488 to 489, reference is made to the dimensions of the project to be evaluated, such as processes and impact. However, I appreciate that another dimension is missing to evaluate, such as the structure, which includes human and material resources to complete the comprehensive evaluation of the project. Therefore, I recommend that, to enrich the project, you include this parameter included by Avedis Donabenian in his scientific works, as it is in his article “Evaluating the Quality of Medical Care.”
